# Chitosan nanoparticles improve the effectivity of miltefosine against *Acanthamoeba*

**Alireza Latifi[1], Fariba Esmaeili[2], Mehdi Mohebali[1,3], Setayesh Yasami-Khiabani[4], Mostafa Rezaeian[1], Mohammad Soleimani[5], Elham Kazemirad[1]*, Amir Amani[6]***

**1** Department of Medical Parasitology and Mycology, School of Public Health, Tehran University of Medical Sciences, Tehran, Iran, **2** Department of Medical Nanotechnology, School of Advanced Technologies in Medicine, Tehran University of Medical Sciences, Tehran, Iran, **3** Center for Research of Endemic Parasites of Iran (CREPI), Tehran University of Medical Sciences, Tehran, Iran, **4** Department of Parasitology, Pasteur Institute of Iran, Tehran, Iran, **5** Department of Ocular Trauma and Emergency, Farabi Eye Hospital, Tehran University of Medical Sciences, Tehran, Iran, **6** Natural products and medicinal plants Research center, North Khorasan University of Medical Sciences, Bojnurd, Iran

* kazemirad@tums.ac.ir, ekazemirad@yahoo.com (EK); aamani@nkums.ac.ir (AA)

## Abstract

### Background

*Acanthamoeba* keratitis (AK) is a corneal sight-threatening infection caused by the free-living amoebae of the genus *Acanthamoeba*. Early and appropriate treatment significantly impacts visual outcomes. Mucoadhesive polymers such as chitosan are a potential strategy to prolong the residence time and bioavailability of the encapsulated drugs in the cornea. Regarding the recent administration of miltefosine (MF) for treating resistant AK, in the present study, we synthesized miltefosine-loaded chitosan nanoparticles (MF-CS-NPs) and evaluated them against *Acanthamoeba*.

### Methodology/Principal findings

Chitosan nanoparticles (CNPs) were prepared using the ionic gelation method with negatively charged tripolyphosphate (TPP). The zeta-potential (ZP) and the particle size of MF-CS-NPs were 21.8±3.2 mV and 46.61±18.16 nm, respectively. The release profile of MF-CS-NPs indicated linearity with sustained drug release. The cytotoxicity of MF-CS-NPs on the *Vero* cell line was 2.67 and 1.64 times lower than free MF at 24 and 48 hours. This formulation exhibited no hemolytic activity *in vitro* and ocular irritation in rabbit eyes. The $IC_{50}$ of MF-CS-NPs showed a significant reduction by 2.06 and 1.69-fold in trophozoites at 24 and 48 hours compared to free MF. Also, the MF-CS-NPs $IC_{50}$ in the cysts form was slightly decreased by 1.26 and 1.21-fold at 24 and 48 hours compared to free MF.

### Conclusions

The MF-CS-NPs were more effective against the trophozoites and cysts than free MF. The nano-chitosan formulation was more effective on trophozoites than the cysts form. MF-CS-NPs reduced toxicity and improved the amoebicidal effect of MF. Nano-chitosan could be

**Data Availability Statement:** The authors confirm that all data underlying the findings are fully available without restriction. All relevant data are

within the paper and its Supporting Information files.

**Funding:** This research has been supported by Tehran University of Medical Sciences and Health Services grant no. 99-2-211-49861. The funders had no role in study design, data collection, and analysis, decision to publish, or preparation of the manuscript.

**Competing interests:** The authors have declared that no competing interests exist.

an ideal carrier that decreases the cytotoxicity of miltefosine. Further analysis in animal settings is needed to evaluate this nano-formulation for clinical ocular drug delivery.

## Author summary

*Acanthamoeba* keratitis (AK) is a painful corneal disease that causes vision loss if not treated promptly. AK incidence is increasing worldwide, especially among those who wear contact lenses. Prompt and proper treatment is essential for complete recovery of vision. The successful treatment has been complicated due to low efficacy, toxicity, and ineffectual ocular drug delivery. Mucoadhesive polymers, like chitosan nanoparticles, are a promising approach to enhancing drug residence time and transcorneal permeation. Miltefosine (MF) is an effective medication for treating refractory AK. In the present study, miltefosine-loaded chitosan nanoparticles were prepared, and their therapeutic effect and cytotoxicity were compared with free miltefosine. The MF-CS-NPs demonstrated a significant decrease in the viability of trophozoite forms compared to free miltefosine. The nano-chitosan formulation was more effective toward trophozoites than the cysts form. Overall, the chitosan nanoparticles improved the effectiveness of miltefosine against *Acanthamoeba*. Besides, this formulation notably reduced toxicity compared to free MF, exhibiting no *in-vivo* irritation. The nano-chitosan carrier can be proposed as an ideal nanocarrier for future evaluation in AK treatment.

## Introduction

*Acanthamoeba* species are the causative agents of a severe corneal infection designated as *Acanthamoeba* keratitis (AK) [1]. The global annual incidence of AK has been estimated to be 23,561 cases, accounting for 2% of corneal infections. The infection affects immunocompetent individuals following contact lens misuse or corneal trauma [2]. In the past two decades, increased use of contact lenses, along with mishandling and poor contact lens hygiene, have elevated microbial keratitis, especially *Acanthamoeba* keratitis [3].

In the early stages, *Acanthamoeba* invades the anterior cornea and, during the later stage, gradually penetrates the deeper corneal tissue, leading to ulceration, blurred vision, and eventually blindness [4]. Early diagnosis and accurate treatment reduce the risk of treatment failure, long-term visual sequelae, and poor visual outcomes. Nevertheless, the clinical management of AK remains challenging due to the absence of standardized diagnostic tests, clinician awareness, and the potential for treatment failure [5]. The treatment can become problematic as trophozoites transform into dormant double-walled cysts resistant to drug penetration. Hence, the treatment should be extended even after the clinical resolution of the infection to prevent the relapse by the cysts [6]. Moreover, when amoeba penetrates deeply into the corneal stroma layer in the late stage, the low drug delivery across the corneal barrier makes successful treatment remarkably difficult [4]. Despite long and continuous treatments, therapeutic keratoplasty is needed in some refractory AK cases to restore visual acuity [7].

Polyhexamethylene Biguanide (PHMB) and Chlorhexidine are two medications that are effective at low concentrations [8]. However, they may develop resistance and side effects like cataracts, iris atrophy, and peripheral ulcerative keratitis [9]. In the last decade, miltefosine (hexadecyl phosphocholine) as an alkyl phosphocholine compound has successfully treated protozoal infections, particularly visceral leishmaniasis [10]. Studies have indicated lower cell

toxicity for MF than PHMB and Chlorohexidine [11,12]. Topical MF has shown promising efficacy in treating AK in animal models, and the oral administration of MF has been used to treat resistant AK cases [13].

The ocular administration of drugs comprises various challenges, such as low retention time on the ocular surface, poor bioavailability, and permeability, along with side effects [14]. Nano-based ocular drug delivery offers new formulations with controlled drug release, improved ocular bioavailability, enhanced cornea permeation, and reduced eye irritation [15].

Polymeric nanoparticles have been widely used in drug delivery systems due to their chemical versatility, biocompatibility, and biodegradability [16]. Chitosan is a nontoxic polymer derived from chitin, primarily found in crustacean shells. It is produced through the N-deacetylation of chitin and exhibits similarities to cellulose [17]. Chitosan has various pharmacological properties such as immune-potentiation, antioxidant, and antibacterial activities [18]. Chitosan nanoparticles (CS-NPs) have been employed as a drug delivery system to treat *Plasmodium vivax* and *Leishmania* [19,20]. As a mucoadhesive polymer, chitosan is a potential carrier to prolong the residence time and bioavailability of the encapsulated drugs on the ocular surfaces and enhance intraocular penetration. It can adhere to the eye and reduce the drug drainage rate due to its viscosity [21,22]. Considering the advantages of chitosan polymer and successful treatment of AK with miltefosine, here we evaluated a new nano-formulation of miltefosine-loaded chitosan nanoparticles (MF-CS-NPs) against trophozoites and cysts of *Acanthamoeba* genotype T4, isolated from keratitis case.

## Material and methods

### Ethics statement

The protocol of the present experimental study was approved by the Ethics Committees of the Tehran University of Medical Sciences, Iran (No. IR.TUMS.SPH.REC.1399.285). The experiments on animals were performed according to the School of Public Health considerations for ethical care and use of animals following the guidelines of ICLAS (International Council for Laboratory Animal Science).

### Compounds

MF was purchased from Selleckchem company (USA); chitosan (MW = 100 kDa, DD = 93%) and sodium tripolyphosphate (TPP) were obtained from Easter Holding Group (China). Fetal bovine serum, Penicillin/Streptomycin, and RPMI were purchased from Gibco (Gibco, USA). DMSO, acetic acid, were purchased from Merck Chemicals, and 3-(4,5-dimethyl- 2-thiazolyl)- 2,5-diphenyl-2H-tetrazolium bromide (MTT), and Chlorhexidine from Sigma-Aldrich (Hamburg, Germany).

### Preparation of nanoparticles

Chitosan nanoparticles were prepared according to the ionotropic gelation process [23]. Chitosan solutions with concentrations of 0.5 wt% were dissolved in 1% acetic acid. The mixture was stirred for 24 hours to obtain a perfectly transparent solution. CS-NPs were obtained upon adding 1 mL TPP aqueous solution (1 mg/mL) to 5 mL chitosan solution (2 mg/mL) and continuously stirred (400 rpm) at room temperature. To obtain MF-CS-NPs, 1 mL MF (5 mg/mL) was added to 1 mL TPP (1 mg/mL), and the solution was added dropwise into 5 mL chitosan solution (2 mg/mL) under constant magnetic stirring at 400 rpm, and room temperature. The non-entrapped drug was removed by centrifugation at 15,000g for 45 min at 4°C, and the pellet was resuspended in pure water. The purified nanoparticles were freeze-dried for further characterization (Fig 1).

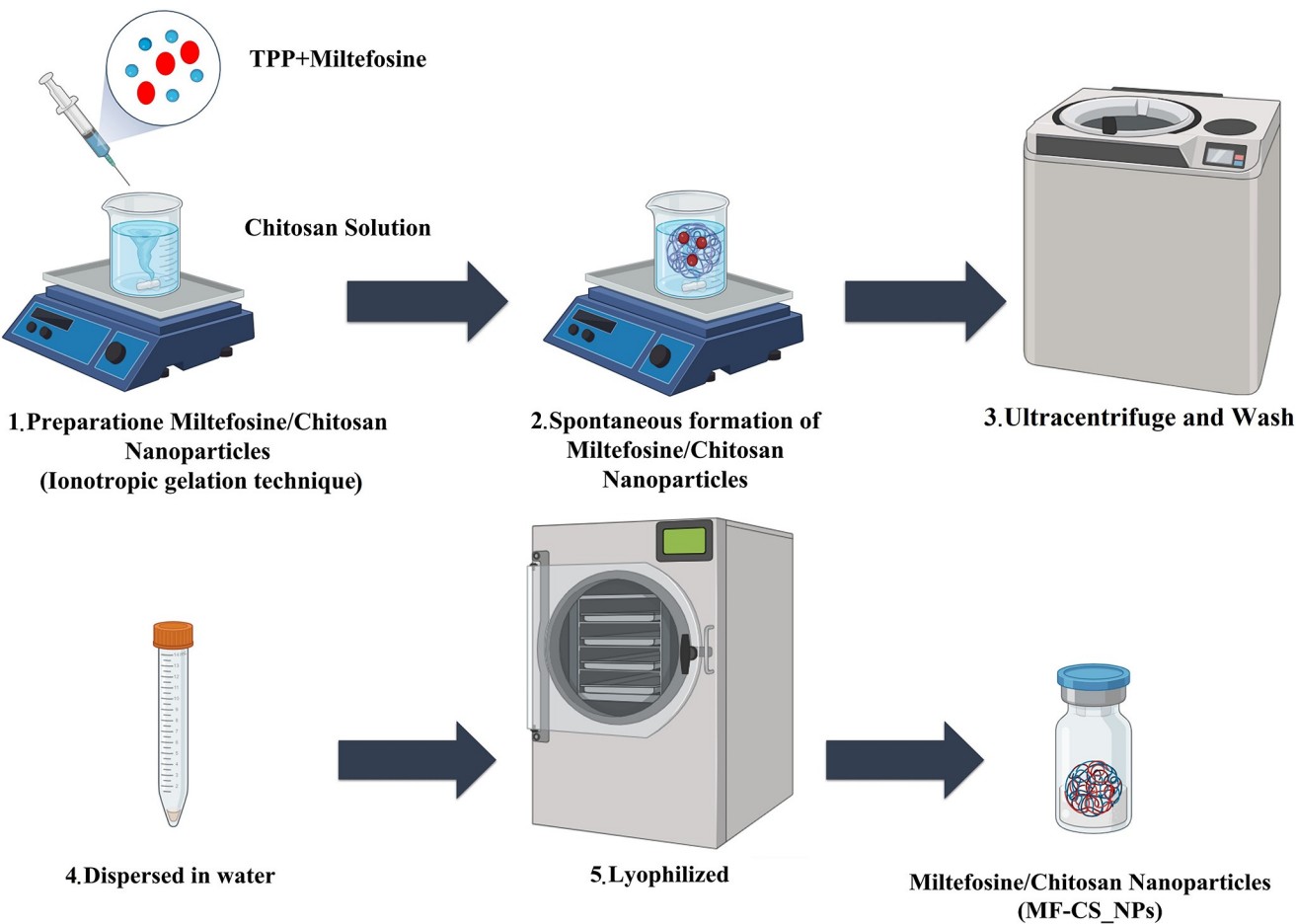

**Fig 1. Schematic illustration of the preparation of MF-loaded chitosan nanoparticles by ionotropic gelation method.** The illustration was generated using Biorender.com (Full license).

## Characterization of MF-CS-NPs

The size and zeta potential of the nanoparticles were obtained by photon correlation spectroscopy using nano Zetasizer (Malvern Zetasizer, UK). The samples were prepared with deionized water at appropriate concentrations. The surface morphology of the MF-CS-NPs was observed using Field Emission Scanning Electron Microscopy (FESEM). The nanoparticle suspensions were spread on a glass plate and dried at room temperature. The dried nanoparticles were coated with gold metal under vacuum and then examined by FESEM (TESCAN, MIRA3, Czech Republic). Also, the size and morphology of the MF-CS-NPs were observed by transmission electron microscopy (TEM) (Philips CM200 EFG, FEI Company, Eindhoven, Netherlands). The chemical structure and complex formation of MF-CS-NPs were analyzed by FTIR (Nicolet, 5DX/550II, USA). The samples were prepared by grinding the dry specimens with potassium bromide (KBr) and pressing the mixed powder to form disks, followed by examining through FTIR.

## Evaluation of drug encapsulation

MF-CS-NPs were prepared from MF (5 mg/mL, 1 mL), TPP (1 mg/mL, 1 mL), and chitosan (2 mg/mL, 5 mL). The solution was pelleted by centrifugation at 15,000g for 45 min at 4°C, and

then the concentration of nonencapsulated MF was determined in the supernatant. Since MF has no UV absorbance, the MF concentration was measured by a colorimetric assay based on the complexation of zwitterionic MF with anionic ammonium ferric thiocyanate ($NH_4Fe$ $[NCS]_4$) dye to form a colored complex with ammonium ferric thiocyanate, which can be extracted in organic solvents. This procedure was modified for MF, and 1, 2-dichloroethane was used to extract the brown-red-colored complex. The colored complex visually and spectrophotometrically was assessed at 460 nm [24]. The calibration curve was plotted using concentrations of 2, 4, 8, 16, and 32 μg/mL MF based on the miltefosine complexation with ammonium iron thiocyanate dye (S1A and S1B Fig). The non-trapped MF was estimated in the supernatant medium. Finally, the following equations were used to calculate the encapsulation efficiency (EE%) and loading efficiency (LE%):

$$EE(\%) = \frac{Total\ MF - Free\ MF}{Total\ MF} \times 100$$

$$LE(\%) = \frac{Total\ MF - Free\ MF}{Weight\ of\ Nanoparticles} \times 100$$

### In vitro drug release assay

The MF-CS-NPs (30 mg) were dispersed in 5 mL PBS (pH 7.4) and transferred into the dialysis tubing (MWCO:12,000 Da; Sigma-Aldrich). The tubing was moved into 30 mL PBS at 37˚C under constant shaking at 100 rpm. Sampling was performed at 0, 2, 4, 6, 8, 10, 12, 14, 16, 18, 20, and 30 h, and the medium was replaced with fresh PBS. Triplicate samples were analyzed for each time point. The concentration of MF released in PBS was determined by a UV spectrophotometer using a colorimetric assay [25].

### Stability analysis

The formulation of MF-CS-NPs was selected for stability studies based on drug content, particle size, zeta potential, and *in vitro* release of MF-CS-NPs. The nanoparticles were synthesized using chitosan 2 mg/mL, TPP 1 mg/mL, and miltefosine 5 mg/mL, packed in the hard gelatin capsules and stored at 25 ± 2˚C, 60% ± 5% RH. The drug content was determined monthly for six months based on the stability studies (ICH Guidelines) [26]. The Zeta potential of MF-CS-NPs was investigated at zero, three, and six months of incubation.

### *Vero* cell lines culture

*In vitro*, cytotoxicity assay was performed on the *Vero* cell line. The cells were maintained in RPMI 1640 (Roswell Park Memorial Institute) supplemented with 10% heat-inactivated fetal bovine serum (FBS), 100 U/mL penicillin, and 100 μg/mL streptomycin (Gibco, Waltham, MA, USA). The cells were incubated at 37˚C in a humidified atmosphere of 95% air and 5% $CO_2$ for 24 hours, and then the adherent cells were detached using a 0.25% trypsin-EDTA solution. The cells were counted using a Neubauer chamber, and cell viability was determined by the trypan blue dye exclusion method.

### Cytotoxicity determination

The cytotoxicity of MF-CS-NPs on *Vero* cell lines was determined using an MTT assay. The cells ($2 \times 10^4$) were seeded in triplicate in 96 well microplates and incubated in 5% $CO_2$ at 37˚C for 24 h. The cells were exposed to a graded concentration of 9.8, 19.6, 39.2, 78.125, 156.25, 312.5, and 625 μg/mL and incubated for 24 and 48 h. The untreated cell was used as a

negative control. Then, the media was removed from all the treated cells, replaced with new media containing MTT (0.5 mg/mL), and reincubated for three hours at 37˚C. MTT solution was removed, and 100 μL of DMSO was added to dissolve insoluble formazan crystals within viable cells' mitochondria. The plate was incubated with DMSO for 5 min with gentle shaking. The cell viability was determined by measuring the absorbance on a microplate reader (Microplate Reader, BioTek, USA) at λmax 570 nm. The half-maximal inhibitory concentration ($IC_{50}$) was calculated by plotting the dose-response curve.

## Biocompatibility assay of MF-CS-NPs

The biocompatibility assay was conducted as previously described [27]. Briefly, in a test tube containing 3.2% sodium citrate, red blood cells (RBCs) were collected by centrifugation (1500 rpm, 10 min) of a human blood sample. The RBCs were washed three times with NaCl (0.9% w/v) and diluted to prepare a stock dispersion. Then, 50 μL of the stock dispersion was mixed with either 950 μL of MF-CS-NPs or 230 μg/mL of MF. The mixture was incubated at 37˚C for one hour and centrifuged at 10,000 rpm for five minutes. The hemolysis rate was assessed by measuring the absorbance of the supernatant at 540 nm against the negative control (PBS solution) and positive control (0.1% Triton-X-100). Finally, the hemolysis rate was calculated according to the following equation:

$$\text{Hemolysis rate}\% = \frac{\text{Absorbance of Test} - \text{Absorbance of Negative control}}{\text{Absorbance of Positive control} - \text{Absorbance of Negative control}} \times 100$$

## *Acanthamoeba* isolate

This study was performed with the *Acanthamoeba* genotype T4, the most prevalent in clinical and environmental samples and the most virulent with the highest binding potential to cornea cells [28]. This isolate was derived from a patient suffering from *Acanthamoeba* keratitis in December 2020, and after determining the genotype, it was kept axenically (Acc. No. MT820305) [29].

## *Acanthamoeba* cultivation

Axenic cultures were obtained as described previously [30]. Briefly, cysts were removed from non-nutrient agar (NNA) plates using cell scrapers, washed by centrifugation three times with saline, and counted with the Neubauer chamber slide. A 3% HCl solution was added to the cysts and incubated at room temperature overnight to inactivate the bacteria. The cysts were washed with PBS, transferred into a protease-peptone-yeast extract-glucose medium (PYG), and incubated at 30˚C. After 72 hours, actively growing trophozoites were harvested by centrifugation at 500g for 7 minutes. For preparing the mature cyst, trophozoite was cultured on 1.5% NNA with 5 μL of heat-killed *Escherichia coli* and incubated at 30˚C for three weeks. The cysts were harvested, washed in PBS, and then treated with 0.5% sodium dodecyl sulfate (SDS) to lyse non-mature cysts. The trophozoite and cysts were then counted using a Neubauer hemocytometer, and $10^5$ cells/mL were adjusted for in vitro assay.

## In vitro assay

Experiments were conducted in microtiter plates with 24 wells at 37˚C under sterile conditions as described previously [31]. One hundred-five cells/mL of trophozoites or cysts were used for the drug assay. MF, MF-CS-NPs, and CS-NPs were tested against trophozoites at a concentration of 9.8, 19.6, 39.2, 78.125, 156.25, 312.5, and 625 μg/mL, and cysts at a concentration of 156.25, 312.5, 625, 1250, 2500 and 5000 μg/mL. Chlorhexidine (0.02%) and untreated

trophozoites or cysts were used as positive and negative controls, respectively [32]. The reduction in amoeba cells was determined by counting the viable cells with a Neubauer hemocytometer after 24 and 48 hours. Trypan blue was served as the viability indicator. All the experiments were conducted in two independent experiments, each time in triplicate.

### Flow cytometry

The cysts were exposed to a 50% inhibitory concentration of MF and MF-CS-NPs for 24 hours and analyzed by flow cytometry to confirm the viability obtained by microscopy. The flow cytometry was not conducted on the trophozoite due to their variable size. The viability of cysts was assessed with modifications to the method described by Khunkitti et al. [33]. Briefly, $10^5$ cysts/mL were incubated with 50% inhibitory concentration ($IC_{50}$) of drugs and then stained with propidium iodide (PI). Flow cytometric analyses were performed with a FACSCalibur fluorescence-activated cell sorter system (Becton Dickinson, Heidelberg, Germany). Illumination was from a 15-mW, 488-nm argon-ion laser. Nonviable cells were stained with PI fluoresced red with a 585/42 filter (FL2-H). Heat-killed cysts at 90°C for 20 min that fluoresced red with PI stain served as a control for the staining procedure. Suspensions containing viable untreated cysts served as controls. Control amoeba populations that excluded PI (viable cells) were used to select analysis gates in FL2-H versus FSC-H (forward scattered light measure of cell size) dot plots. The dot plots were also gated by forward scatter to eliminate analysis of noise and smaller particles (<3 μm in diameter). Treated amoebae cysts were superimposed into the previously selected analysis gates. The viability percentage was analyzed based on at least 10,000 cysts.

### Eye irritancy evaluation

The potential ocular irritancy and harmful effects of MF-CS-NPs eye drop dispersion were determined based on a modified scoring system for ocular irritation testing, according to guidelines of the Organization for Economic Cooperation and Development (OECD) [34]. The eye irritancy was evaluated in six male domestic rabbits weighing 1.5 and 2.5 kg. The animals were divided into two groups of three. Group I received a blank MF-CS-NPs eye drop solution with a concentration of 1023 μg/mL; group II received a 0.9% NaCl drop solution. The right eye of each rabbit was treated with two drops of the assigned treatment twice daily for 72 hours. The animals were monitored for ocular conditions such as discomfort and clinical signs in the conjunctiva, cornea, and eyelids.

### Statistical analysis

The data were presented as the mean ± standard deviation (SD). Data analysis was performed using GraphPad Prism version 9.5.0 with a two-tailed Student's t-test and one-way analysis of variance (ANOVA). A *p* value < 0.05 was considered significant.

## Results

### Characterization and morphology of MF-CS-NPS

The size and surface morphology of MF-CS-NPs were determined using dynamic light scattering (DLS), FESEM, and TEM. The average diameters of the nanoparticles were 53.28±15.13 nm using FESEM (Figs 2 and 3). As shown in Fig 2, the MF-CS-NPs displayed a spherical shape with irregular surface morphology. The average particle size of MF-CS-NPs was 46.61 ±18.16 nm, determined by the TEM micrograph (Fig 3). The drug loading content of MF in

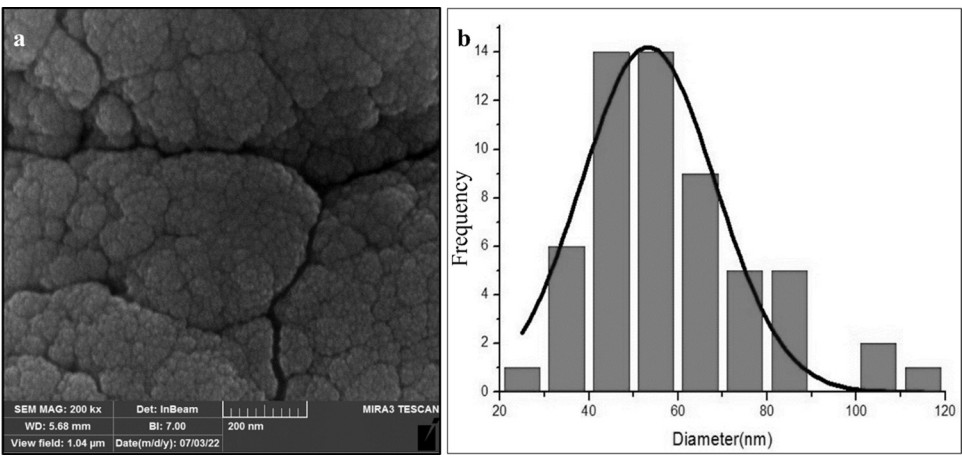

**Fig 2.** Field Emission Scanning Electron Microscopy (FESEM) analysis of miltefosine-loaded chitosan nanoparticles (a), The particle size histogram of MF-CS-NPs (b).

nanoparticles was 83.3±4.3 μg/mL. The MF-CS-NPs possessed a positive zeta potential of 21.8 ±3.2 mV (Fig 4 and Table 1).

## Studying loading of MF-CS-NPs by FTIR spectrum

The FTIR results revealed the structural properties of MF, CS-NPs, and MF-CS-NPs (Fig 5). It demonstrated the differences between unloaded and loaded nanoparticles. In FTIR analysis of drug carriers, a high double peak (2917 and 2850 cm$^{-1}$) was detected, corresponding to $CH_2$ stretching in the long hexadecyl chain that indicateed MF in drug carriers [24].

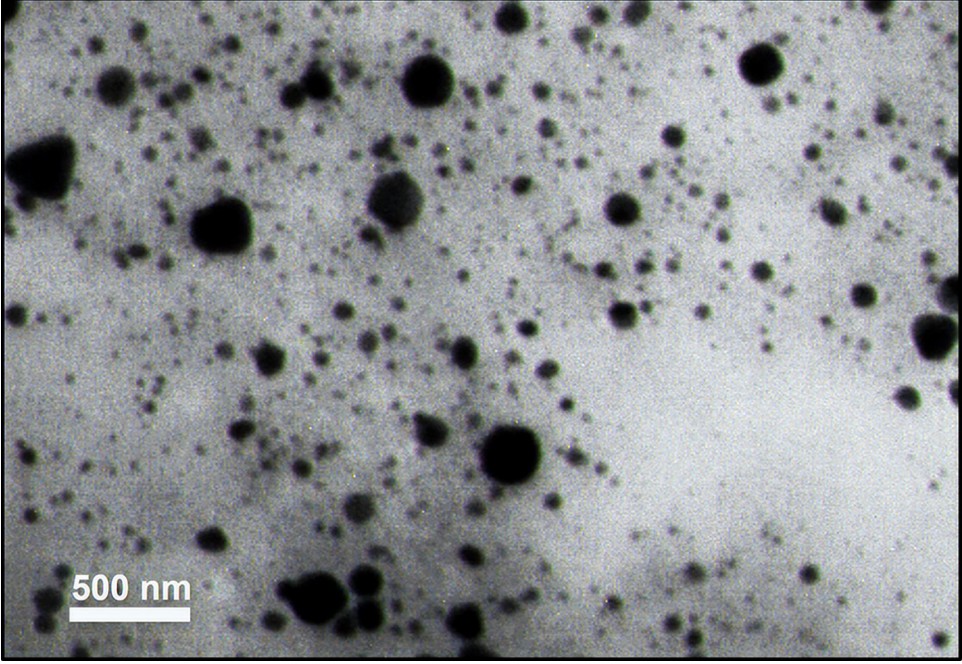

**Fig 3. Transmission electron microscopy (TEM) image of the MF-CS-NPs.**

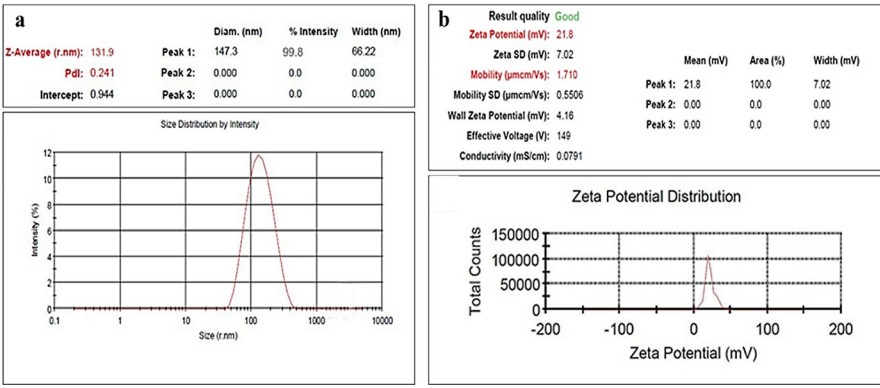

**Fig 4. Physical characterization of nanoparticles.** (a) Particle size distribution of miltefosine-loaded chitosan nanoparticles. The X-axis is the size distribution of particles, and the Y-axis is the number of particles. (b) The zeta potential of miltefosine-loaded chitosan nanoparticles.

## In vitro release study

The in-vitro release for MF and MF-CS-NPs was conducted in phosphate buffer (pH 7.2). Fig 6 shows that 46.7 ± 5.27% of the total loaded MF was released during the first four hours. The total cumulative MF percent released from nanoparticles after 30 h of incubation was 83.2 ± 1.8%. The findings show that chitosan nanoparticles provide sustained MF release.

## Stability analysis result

The drug content slightly decreased during six months of storage at 25 ± 2˚C, 60% ± 5% RH (Fig 7). After six months, the drug content was more than 60%, demonstrating the acceptable stability of miltefosine in chitosan nanoparticles. The Zeta potential of MF-CS-NPs following zero, three-, and six-months incubation in accelerated stability test conditions was 21.8±3.2, 20.8±2.2, and 19.8±4.3 mV, respectively. The slight decrease in zeta potential may be attributed to nanoparticle aggregation and morphological changes.

## Cell cytotoxicity assay of MF, MF-CS-NPs, and CS-NPs

The cytotoxicity of MF-CS-NPs on *Vero* cells was evaluated. The $IC_{50}$ value of free MF was 67.55±8.16 and 33.72±3.35 μg/mL after 24 and 48 h incubation, whereas the values for MF-CS-NPS were 180.5±15.00 and 55.15±4.70 μg/mL after 24 and 48 h, respectively. The results showed that the cytotoxicity of MF-CS-NPS significantly decreased by 2.67-fold compared to free MF at 24 hours ($p<0.05$) (Fig 8). After 48 hours, due to the drug release, MF-CS-NPS exhibited a slight reduction by 1.64-fold in cytotoxicity compared to the free drug. Formulated CS-NPs were not toxic in the investigated concentrations (Fig 8 and Table 2).

**Table 1. Physicochemical properties of miltefosine-loaded chitosan nanoparticles.**

| Parameters (MF-CS-NPS) | Amount |
|---|---|
| **Encapsulation Efficiency (EE)** | 86.33±4.3% |
| **Loading Content (LE)** | 30.17±1.4% |
| **Average Size (DLS)** | 131.9±10.3 nm |
| **Zeta Potential** | 21.8±3.2 mV |
| **Average Size (FESEM)** | 53.28±15.13 nm |
| **Average Size (TEM)** | 46.61±18.16 nm |

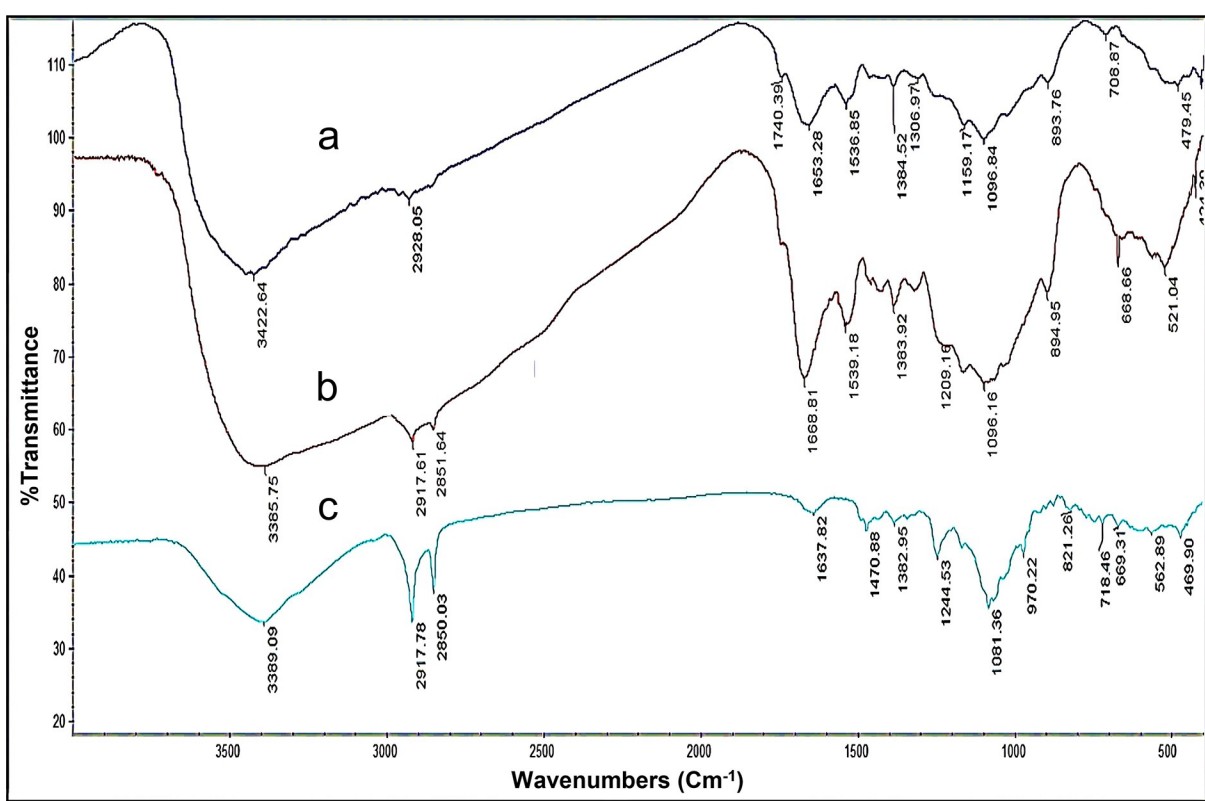

**Fig 5.** Fourier transforms infrared spectra of nano chitosan, (a) miltefosine-loaded chitosan nanoparticles (b), and miltefosine (c).

### Hemolysis assay

The hemolysis percentage of the MF-CS-NPs and MF was 2.1 and 7.4% at the 230 μg/mL concentration of MF after 2 h incubation, indicating non-hemolytic and good hemolysis protection activity. According to the ISO/TR 7406 guideline, a hemolysis rate lower than 5% is considered non-hemolytic, which is required for materials to be used in biomedical applications [35].

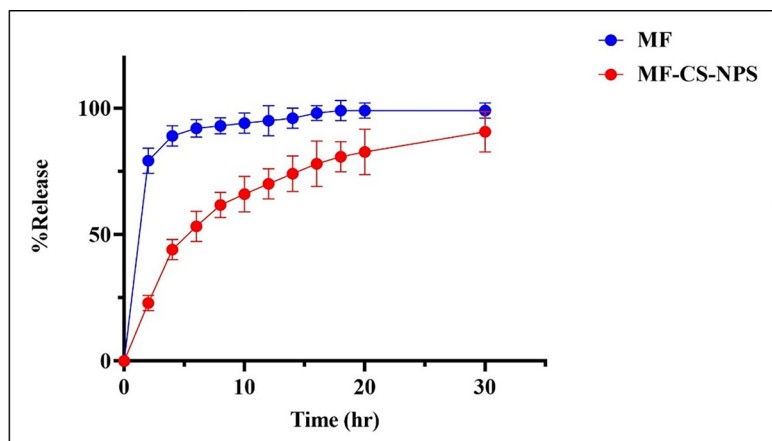

**Fig 6. In vitro drug release profile of MF and miltefosine-loaded chitosan nanoparticles in phosphate buffer (pH 7.2).**

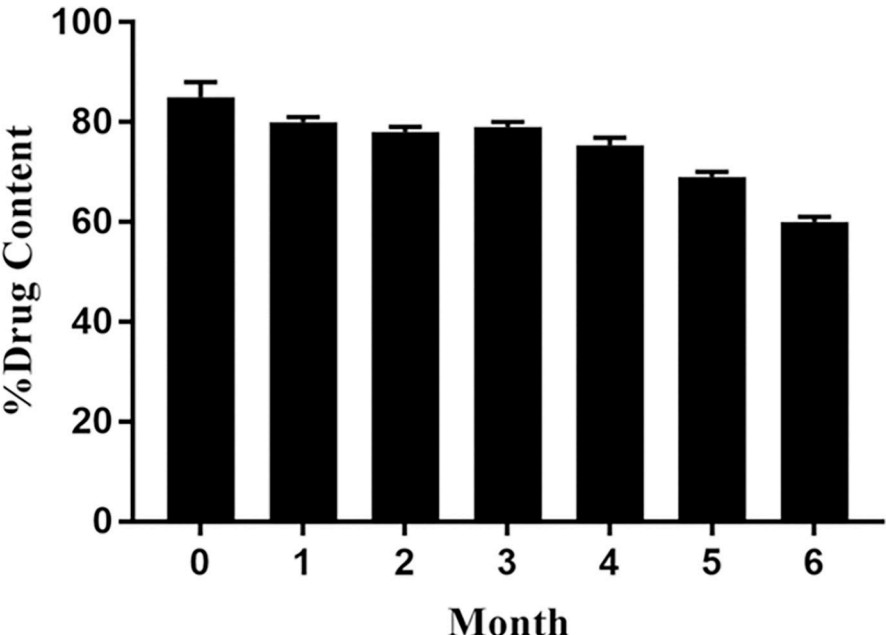

**Fig 7. Percentage drug content of MF-CS-NPs stored at 25 ± 2°C, 65% ± 5% RH after 0–6 months storage.**

## Effects of MF, MF-CS-NPs, and CS-NPs on trophozoites and cysts

The activity of MF-CS-NPs, MF, and CS-NPs against trophozoites and cysts was evaluated using trypan blue staining at 24 and 48 h. The effectivity of MF-CS-NPs and MF was higher in the trophozoite compared to the cyst form. The MF-CS-NPs and MF significantly decreased the trophozoite viability ($p<0.05$) at 24 and 48 hours (Fig 9). At 24 hours, the $IC_{50}$ for MF-CS-NPs and MF against the trophozoite was 93.57±5.00 and 192.8±16.30 μg/mL, respectively. The $IC_{50}$ of MF-CS-NPs showed a significant 2.06-fold reduction in trophozoite at 24 hours compared to free MF. After 48 h, the $IC_{50}$ for MF-CS-NPs was slightly lower by 1.69-fold than MF due to the release of miltefosine from the chitosan by increasing exposure time.

MF-CS-NPs were more effective on cysts than free miltefosine, and the $IC_{50}$ of MF-CS-NPs in the cyst form was slightly decreased by 1.26 and 1.21-fold at 24 and 48 hours compared to free MF. CS-NPs were ineffective in the investigated concentrations. The $IC_{50}$ values for

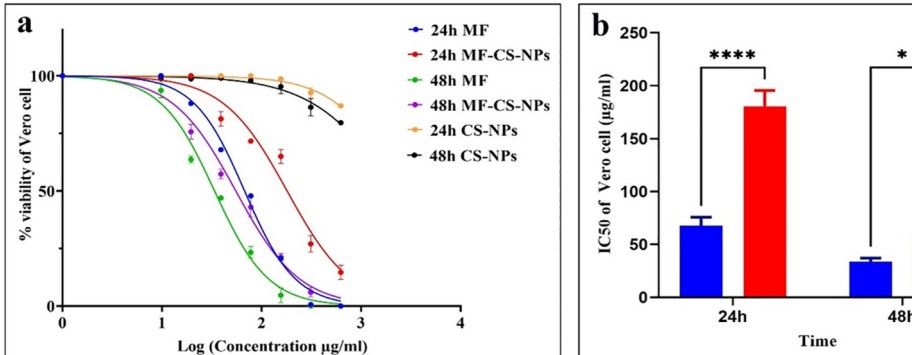

**Fig 8. Comparison of cell cytotoxicity effect.** The viability of *Vero* cell after treatment with MF, MF-CS-NPs, and CS-NPs (24–48 h) (a). The $IC_{50}$ of MF and MF-CS-NPs against *Vero* cell line ($p<0.05$) (b). (****, $p<0.0001$; ns: non-significant).

**Table 2. Effects of MF, MF-CS-NPs, and CS-NPs on *Vero* cell viability after 24 and 48 hours of drug exposure.**

|  | IC$_{50}$ (24h) µg/mL | 95% CI | IC$_{50}$ (48h) µg/mL | 95% CI |
|---|---|---|---|---|
| **MF** | 67.55±8.16 | 63.16 to 72.23 | 33.72±3.35 | 31.01 to 36.66 |
| **MF-CS-NPs** | 180.5±15.00 | 159.1 to 204.8 | 55.15±4.70 | 49.56 to 61.36 |
| **CS-NPs** | 2534±386.70 | 1958 to 3493 | 2242±357.60 | 1692 to 3253 |

CS-NPs were much higher and did not significantly decrease trophozoite and cyst viability ($p>0.05$) at 24 and 48 hours in comparison with MF-CS-NPs and MF compounds (Fig 9 and Table 3). The 0.02% chlorhexidine as a positive control resulted in 100% trophozoites and cysts death after 24 and 48 hours.

## Flow cytometry result

The flow cytometry experiment confirmed the cell viability results of the microscopy (Figs 10 and 11). The cysts were exposed to 50% MF and MF-CS-NPs inhibitory concentration for 24 hours and stained with propidium iodide. The flow cytometry assay confirmed the trypan blue staining results, and the cyst death percentage following exposure to MF and MF-CS-NPs was 43.9 and 54.4%, respectively.

## Eye irritancy assessment

The irritation tendency of the MF-CS-NPs solution and 0.9% NaCl eye drop dispersions was evaluated on the rabbit eyes. The MF-CS-NPs and 0.9% NaCl solution showed no changes in

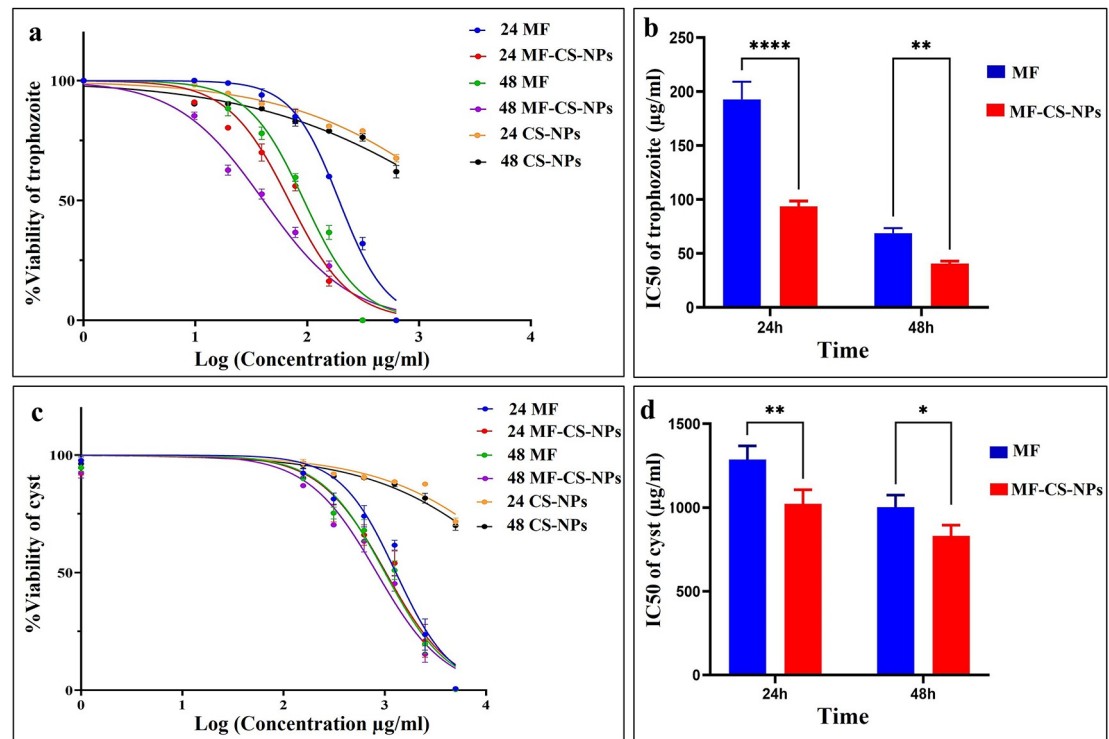

**Fig 9. The in-vitro drug assay against the trophozoites and cysts of *Acanthamoeba*.** The viability percentage of *Acanthamoeba* trophozoite when exposed to MF, MF-CS-NPs, and CS-NPs (24–48 h) (a). The IC$_{50}$ diagram of MF and MF-CS-NPs against *Acanthamoeba* trophozoite (24–48 h) (b). The viability of *Acanthamoeba* cyst form in exposure to MF, MF-CS-NPs, and CS-NPs (24–48 h) (c). The IC$_{50}$ diagram of MF and MF-CS-NPs against *Acanthamoeba* cysts (24–48 h) (d). (****, $p<0.0001$; **, $p<0.01$; *, $p<0.05$).

**Table 3. IC$_{50}$ of MF, MF-CS-NPs, CS-NPs on Trophozoites and cyst form of *Acanthamoeba* after 24 and 48 hours of drug exposure.**

| | Trophozoite | | | | Cyst | | | |
|---|---|---|---|---|---|---|---|---|
| Incubation time | 24h | | 48h | | 24h | | 48h | |
| IC50 | IC$_{50}$ µg/mL | 95% CI | IC$_{50}$ µg/mL | 95% CI | IC$_{50}$ µg/mL | 95% CI | IC$_{50}$ µg/mL | 95% CI |
| MF | 192.8±16.30 | 180.0 to 206.4 | 68.81±4.70 | 60.13 to 78.45 | 1287±81.56 | 1099 to 1499 | 1003±72.01 | 859.6 to 1166 |
| MF-CS-NPs | 93.57±5.00 | 83.42 to 104.7 | 40.62±2.40 | 35.68 to 46.16 | 1023±83.79 | 857.1 to 1215 | 831.8±63.15 | 706.1 to 975.8 |
| CS-NPs | 2301±28.00 | 1830 to 3019 | 2207±398.80 | 1578 to 3385 | 25706±7360 | 15107 to 58203 | 21329±4370 | 14532 to 36270 |

the cornea, iris, and conjunctiva with no redness or secretions. The total irritancy score was zero, suggesting the selected MF-CS-NPs can be considered a safe formulation for ophthalmic application (Fig 12).

## Discussion

The treatment of *Acanthamoeba* keratitis is becoming a severe challenge in managing the disease. Considering the low effectiveness, poor ocular penetration and bioavailability, and toxicity of current topical therapeutic agents, developing more effective and safe therapeutic regimens is essential for treating AK [36]. In the present study, chitosan nanoparticles were employed for loading the miltefosine to evaluate the therapeutic and cytotoxicity effects compared to free miltefosine. MF-CS-NPs were prepared by ionic gelation and showed a high encapsulation efficiency of 86.3% and particle size of 46.61±18.16 nm. The positive zeta potential was 21.8±3.2 mV, related to amino groups in the chitosan structure, enabling the

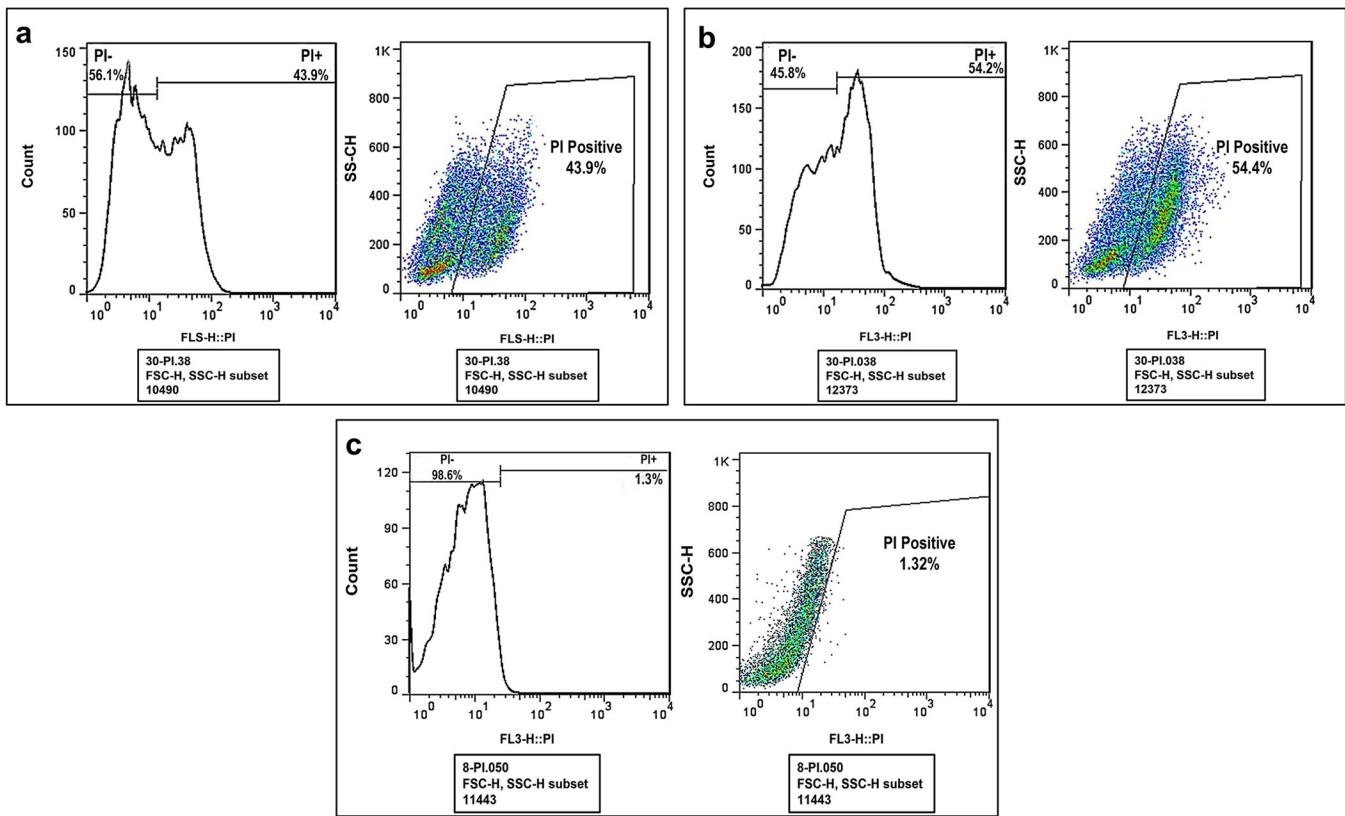

**Fig 10. Flow cytometry diagram of the viability of *Acanthamoeba* cysts**, treated with MF (a) MF-CS-NPs (b) and control (c) after 24 h.

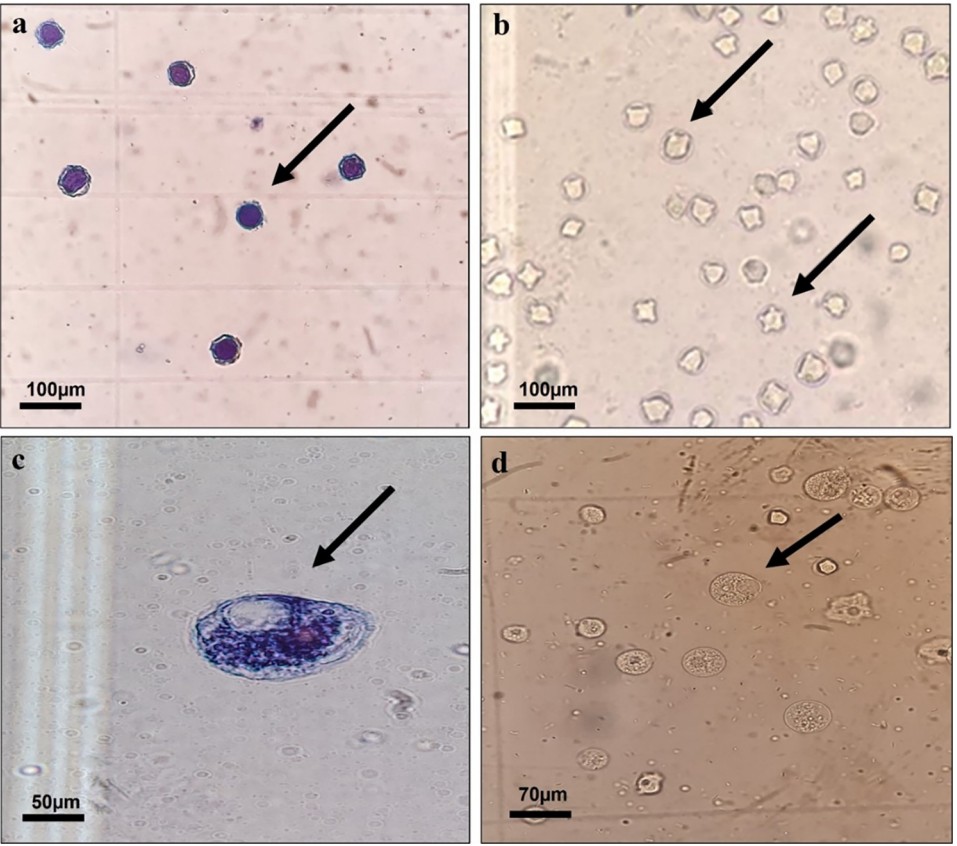

**Fig 11. Microscopic images of *Acanthamoeba* after 24 h of treatment.** (a) *Acanthamoeba* cysts after exposure to MF-CS-NPs, stained with Trypan Blue 0.04%, (b) *Acanthamoeba* cysts before exposure to MF-CS-NPs, (c) *Acanthamoeba* trophozoite after exposure to the MF-CS-NPs, stained with Trypan Blue 0.04%, (d) *Acanthamoeba* trophozoite before exposure to the MF-CS-NPs.

nanoparticles to interact with anionic molecules [37]. The positive zeta potential is essential for ocular drug delivery since it facilitates positive nanoparticle adhesion to the negatively charged cornea surface, prolonging the drug release and enhancing the drug bioavailability in the internal eye tissues [38]. The structural morphology of the NPs showed that MF-CS-NPs exhibit a spherical form with irregular surfaces, which was also previously reported [39]. The small particle size of nanoparticles contributes to mobility and surface interaction, enhancing the antimicrobial activity and the bioavailability of poorly water-soluble molecules [40]. The particle size of MF-CS-NPS was less than 200 nm, which is suitable for ocular drug delivery due to better penetration through the ocular barrier and their low irritation [41].

By monitoring the MF release at pH 7.4, it was found that the chitosan nanoparticles provide sustained drug release, consistent with the previous study [42]. Various factors affect drug release from chitosan nanoparticles, such as polymer swelling, absorbed drug, drug diffusion through the polymeric matrix, polymer erosion or degradation, and a combination [43]. The initial release from the chitosan nanoparticles is most probably due to the release of adsorbed MF. The pH-dependent drug release feature of chitosan nanoparticles is ideal in drug delivery to adjust the MF release rate [44]. In the stability test, after six months of incubation, the drug content was more than 60%, and the zeta potential was 19.8±4.3 mV, indicating the acceptable stability of MF-CS-NPs. During the storage, the drug content and zeta potential slightly

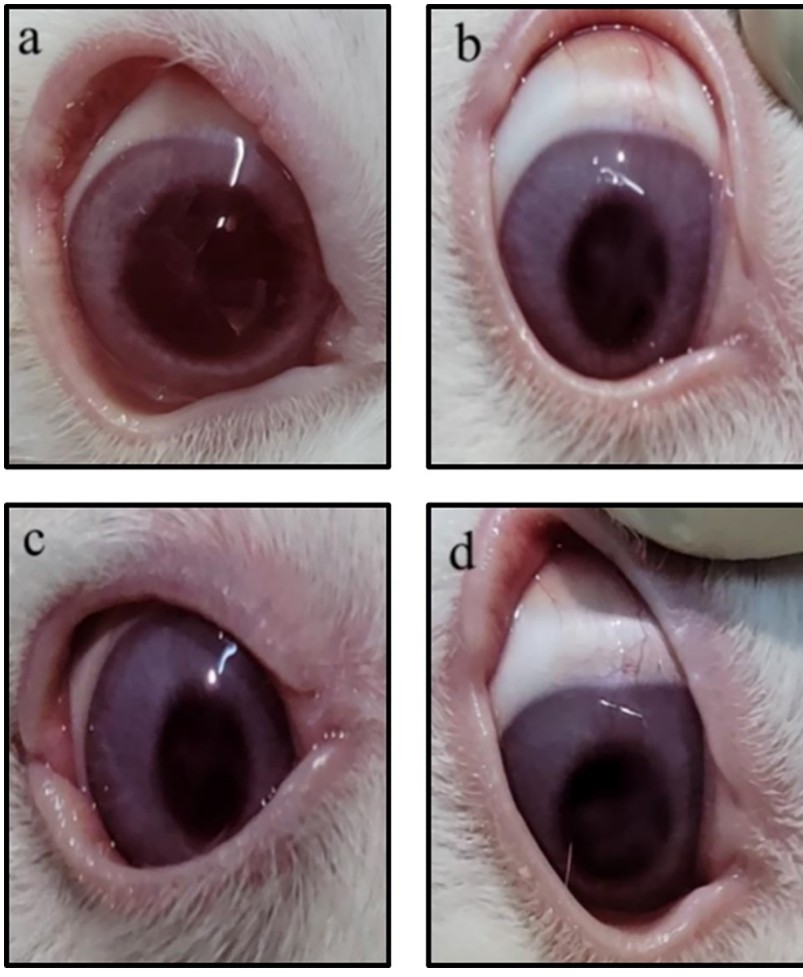

**Fig 12. The irritancy assessment on rabbit eyes.** The NaCl-treated eye at 0 h (a) and 72 h (b); the MF-CS-NPs-treated eye at 0 h (c) and 72 h (d), photographed by the authors.

decreased, possibly due to nanoparticle aggregation and morphological changes. The zeta potential of chitosan-loaded nanoparticles can significantly influence their stability in media through electrostatic repulsion between the particles [45].

In the cytotoxicity assay on the *Vero* cell line, the $IC_{50}$ value of MF-CS-NPs and MF was 180.5±15.00 and 67.55±8.16 µg/mL, respectively, after 24h incubation. The cytotoxicity of MF-CS-NPs was 2.67 times lower than free MF, which might have resulted from the sustained release of MF from nanoparticles, providing minimum contact between MF and the cells [46]. Furthermore, encapsulation can decrease the interaction of the active ingredient with the cells [47]. Several studies reported that chitosan could reduce the cytotoxicity of the drug when used in nanocarrier formulation [48,49]. A survey by Tripathi et al. demonstrated that the chitosan carrier system could reduce the cytotoxicity of amphotericin B and MF, which might be associated with the uptake behavior of nanocarrier systems and slow drug release [50].

Moreover, it was demonstrated that encapsulation of S-nitroso-mercapto succinic acid into chitosan nanoparticles led to decreasing promastigote and amastigote forms of *Leishmania amazonensis* without macrophage toxicity [51]. After 48 hours, the difference between the toxicity of free MF and MF-CS-NPs was not remarkable owing to the drug release (Fig 6). Also,

the free chitosan nanoparticle was nontoxic in the in-vitro assay, which agrees with other surveys [52]. In the current study, MF-CS-NPs showed notably lower hemolytic activity than free MF, representing improved red blood cell compatibility. The lower hemolytic activity is predicted as the drug is entrapped in the nanoparticles [53].

The exact mechanism of MF against *Acanthamoeba* is not entirely understood; nevertheless, its activity could be associated with changes in lipid metabolism, reduction of mitochondrial cytochrome c oxidase, and induction of apoptosis-like cell death in parasites such as *Leishmania* [10]. MF has been utilized in treating amoebic encephalitis cases and against different strains of *Acanthamoeba in vitro* [31,54]. In the present study, drug assay against the trophozoite revealed that MF and MFS-CNPs exhibited $IC_{50}$ values of 192.8±16.30 and 93.57 ±5.00 μg/mL after 24 h and 68.81±4.70 and 40.62±2.40 μg/mL after 48 h incubation, respectively. MFS-CNPs showed increased activity against the trophozoite compared to free MF. This improvement could be attributed to the cellular uptake of the carrier and the slow, sustained release of the drug [50]. Several studies demonstrated that the chitosan nanoparticles accumulate better into the macrophages due to their preferential phagocytosis and are introduced as effective delivery approaches for biopharmaceuticals [50,55]. Indeed, chitosan-coated nanoparticles improved cellular uptake in macrophages compared to uncoated nanoparticles [56]. In addition, the positive charge enables chitosan to attach to the cells efficiently and uptake rapidly [50]. It is assumed that the amoeba trophozoites could phagocyte the chitosan nanoparticles-loaded drug more efficiently than the free drug, which makes it accumulate in the cytoplasm. On the other hand, it is expected that due to the change of pH in the amoeba cytoplasm, compared to the external environment, the drug release faster in the cytoplasm [57].

Our results revealed that in the cyst form, the $IC_{50}$ values for MF and MFS-CNPs were 1287 ±81.56 and 1023±83.79 μg/mL after 24 h and 1003±72.01 and 831.8±63.15 μg/mL after 48 h incubation, respectively. Although the $IC_{50}$ was lower in MFS-CNPs than the free MF, the difference in the cyst form was not remarkable. It assumed that the positive charge of chitosan nanoparticles allows attachment to the negatively charged cyst wall. Nonetheless, the drug penetration into the double-layered cellulose wall of the cyst is complex and requires more prolonged exposure [58]. Previous studies on different antimicrobial agents revealed that the cysticidal concentration is generally higher than the trophozoicidial concentration; thus, more prolonged drug exposure was needed in in-vitro and clinical cases to vanish the cyst form [59]. In a study by Walochnik et al., miltefosine at the concentration of 40 μM eradicated the trophozoites, while a cysticidal effect was observed at 160 μM [31]. Interestingly, the effective drug concentration varies based on the species, genotype, and thickness of double-walled cysts of *Acanthamoeba* [60]. Therefore, more studies are required to optimize and improve the cysticidal activity of the MF-loaded chitosan nanoparticles.

Our results revealed that chitosan was not toxic in the concentration used for preparing the MF-CS-NPS. The $IC_{50}$ value of chitosan was considerably higher than that of MF-CS-NPs and free MF against the trophozoite and cyst forms of *Acanthamoeba*. Although the antiamoebic activity of chitosan is lower than miltefosine, the encapsulation of miltefosine into chitosan nanocarriers, through changes in drug penetration and release, significantly enhances the effectiveness of MF. The nanoparticle size allows their penetration through tight junctions and increases the surface-to-volume ratio, strongly affecting their release outline [61]. Also, a carrier covering the drug molecule offers protection from environmental and biological barriers [62].

In the present study, the eye irritation of MF-CS-NPs was investigated in rabbits' eyes. Topical application of MF-CS-NPs displays no sign of irritation, redness, or abnormal discharge. Our finding agrees with the previous reports on chitosan-based nanocarriers applied for

topical ocular delivery of dexamethasone, forskolin, and clarithromycin, in which topical chitosan nanoparticles were nonirritant [63–65]. Interestingly, chitosan is a cationic polysaccharide with mucoadhesive properties that can bind to the epithelium, making it a desirable drug carrier for ophthalmic applications. Mucoadhesive drug delivery systems are ideal as they can increase the residence time of the drugs at the site of absorption in tissue, providing sustained drug release and minimizing the degradation of drugs in various body sites [21]. In addition, chitosan can increase drug transfer across corneal barriers, offering a promising strategy to increase drug permeability, which is essential to treat the late stage of *Acanthamoeba* keratitis [22].

In conclusion, miltefosine-loaded chitosan nanoparticles were evaluated against *Acanthamoeba* for the first time. This formulation exhibited the optimal size, zeta potential, and stability. Also, the sustained release profile of MF-CS-NPs was confirmed. It exhibited low cytotoxicity toward the *Vero* cell line compared to free MF, without any hemolytic activity *in vitro* and ocular irritation in rabbit eyes. The MF-CS-NPs showed a significant reduction in trophozoite viability compared to free miltefosine while moderately effective against the cyst. Our results revealed that nano-chitosan could be an ideal career that reduced the cytotoxicity of miltefosine. The development of this nano-formulation of miltefosine opens a new era of effective treatment of *Acanthamoeba*. Further studies on the animal model are required to shed light on the effectiveness of the miltefosine-loaded chitosan nanoparticle in ocular drug delivery systems.

## Supporting information

**S1 Fig. The colorimetric assay for miltefosine based on the colored complex formation with ammonium ferric thiocyanate.** (A) The colored complex results from different concentrations of 0, 2, 4, 8, 16, and 32 µg/mL MF, photographed by the authors (B) the calibration curve of the colorimetric assay.
(TIF)

## Acknowledgments

The authors would like to express their gratitude to Prof. Saied Reza Naddaf, Dr Fatemeh Goodarzi, and Dr Mona Koosha for their kind cooperation. We also thank the members of the workgroup Parasitology at the Department of Medical Parasitology and Mycology, School of Public Health, Tehran University of Medical Sciences, for their technical support.

## Author Contributions

**Conceptualization:** Alireza Latifi, Elham Kazemirad, Amir Amani.

**Formal analysis:** Alireza Latifi, Fariba Esmaeili.

**Funding acquisition:** Elham Kazemirad.

**Investigation:** Alireza Latifi, Fariba Esmaeili, Setayesh Yasami-Khiabani.

**Methodology:** Alireza Latifi, Amir Amani.

**Project administration:** Mostafa Rezaeian, Mohammad Soleimani, Elham Kazemirad.

**Supervision:** Mehdi Mohebali, Elham Kazemirad, Amir Amani.

**Writing – original draft:** Alireza Latifi, Elham Kazemirad.

**Writing – review & editing:** Elham Kazemirad.

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
