## [Decision Letter · Decision Letter 0]

12 Oct 2023

Dear Dr Kazemirad,

Thank you very much for submitting your manuscript "Chitosan nanoparticles improve the effectivity of miltefosine against Acanthamoeba" for consideration at PLOS Neglected Tropical Diseases. As with all papers reviewed by the journal, your manuscript was reviewed by members of the editorial board and by several independent reviewers. The reviewers appreciated the attention to an important topic. Based on the reviews, we are likely to accept this manuscript for publication, providing that you modify the manuscript according to the review recommendations. 

Reviewer 2:

I've completely gone through the manuscript titled "Chitosan nanoparticles improve the effectivity of miltefosine against Acanthamoeba" ". The authors report the effects of miltefosine loaded chitosan nanoparticles against the two forms of Acanthamoaba namely, cultured trophozoites and cyst form. The cytotoxicity and hemolytic effects of the MF-CS-NPs were also determined on Vero cell line and human RBCs, respectively. This is a well-conceived and well-carried-out study that have yielded valuable data. However, before this paper can be published, a number of flaws must properly be taken care of. The authors should have a native English speaker revise the manuscript before resubmitting it in order to correct the annoying spelling, grammatical, syntactic, and stylistic errors. 

General and specific comments for the authors

 Introduction 

• The introduction is not for the authors perform a review about the subject but to clearly show why the study is relevant and how the data obtained will contribute to the management of AK. Most of the references cited in this part are outdated. 

• The authors are advised to pay special emphasis on recent challenges in treatment options of AK. 

Methods 

• How the authors confirm the synthesis of NPs

• How the authors standardized the dosing in the experiments

• The authors are advised to perform TEM microscopy of the MF-CS-NPs

• No information is given about how long the isolate of Acanthamoaba used was obtained. 

• The authors should clearly mention the positive and negative control in their experiments

• How the authors measured the encapsulation efficacy and drug loading content of the MF-CS-NPs 

• Penicillin and streptomycin from which source? company, country and city name

• It would be interesting if the authors could perform the experiments in three incubation periods i-e 24h, 48,72h respectively. 

• In In vitro assay the authors used various concentrations of 9.8, 19.6, 39.2, 78.125, 156.25, 312.5, and 199 625 µg/ml against trophozoites while 156.25, 312.5, 625, 1250, 2500 and 5000 µg/ml for cyst. What was the reason for choosing these concentrations?

Results 

The results are well presented. The readers can easily be able to understand what the figures and tables represents. 

 Fig 8 must be removed as it not giving any scientific information

Discussion 

The discussion must be written more straightforward. Rather than repeating background information in the first alinea that has already been given in the Introduction. The authors should have briefly mentioned the rationale for carrying out the study and give a short version of their findings, and then continue in the subsequent alineas to discuss each finding. 

Conclusion 

Conclusion must be informative. The authors should mentioned their results in this section.

Sincerely,

Aiman Abu Ammar

Guest Editor

Charles Jaffe

Section Editor

Reviewer 2:

I've completely gone through the manuscript titled "Chitosan nanoparticles improve the effectivity of miltefosine against Acanthamoeba" ". The authors report the effects of miltefosine loaded chitosan nanoparticles against the two forms of Acanthamoaba namely, cultured trophozoites and cyst form. The cytotoxicity and hemolytic effects of the MF-CS-NPs were also determined on Vero cell line and human RBCs, respectively. This is a well-conceived and well-carried-out study that have yielded valuable data. However, before this paper can be published, a number of flaws must properly be taken care of. The authors should have a native English speaker revise the manuscript before resubmitting it in order to correct the annoying spelling, grammatical, syntactic, and stylistic errors. 

General and specific comments for the authors

 Introduction 

• The introduction is not for the authors perform a review about the subject but to clearly show why the study is relevant and how the data obtained will contribute to the management of AK. Most of the references cited in this part are outdated. 

• The authors are advised to pay special emphasis on recent challenges in treatment options of AK. 

Methods 

• How the authors confirm the synthesis of NPs

• How the authors standardized the dosing in the experiments

• The authors are advised to perform TEM microscopy of the MF-CS-NPs

• No information is given about how long the isolate of Acanthamoaba used was obtained. 

• The authors should clearly mention the positive and negative control in their experiments

• How the authors measured the encapsulation efficacy and drug loading content of the MF-CS-NPs 

• Penicillin and streptomycin from which source? company, country and city name

• It would be interesting if the authors could perform the experiments in three incubation periods i-e 24h, 48,72h respectively. 

• In In vitro assay the authors used various concentrations of 9.8, 19.6, 39.2, 78.125, 156.25, 312.5, and 199 625 µg/ml against trophozoites while 156.25, 312.5, 625, 1250, 2500 and 5000 µg/ml for cyst. What was the reason for choosing these concentrations?

Results 

The results are well presented. The readers can easily be able to understand what the figures and tables represents. 

 Fig 8 must be removed as it not giving any scientific information

Discussion 

The discussion must be written more straightforward. Rather than repeating background information in the first alinea that has already been given in the Introduction. The authors should have briefly mentioned the rationale for carrying out the study and give a short version of their findings, and then continue in the subsequent alineas to discuss each finding. 

Conclusion 

Conclusion must be informative. The authors should mentioned their results in this section.

Reviewer's Responses to Questions

**Key Review Criteria Required for Acceptance?**

**Methods**

-Are the objectives of the study clearly articulated with a clear testable hypothesis stated?

-Is the study design appropriate to address the stated objectives?

-Is the population clearly described and appropriate for the hypothesis being tested?

-Is the sample size sufficient to ensure adequate power to address the hypothesis being tested?

-Were correct statistical analysis used to support conclusions?

-Are there concerns about ethical or regulatory requirements being met?

Reviewer #1: Yes, the objectives of the study is clearly articulated with a clear testable hypothesis stated and the study design is appropriate to address the stated objectives.

Reviewer #2: (No Response)

**Results**

-Does the analysis presented match the analysis plan?

-Are the results clearly and completely presented?

-Are the figures (Tables, Images) of sufficient quality for clarity?

Reviewer #1: Yes, the results are clearly and completely presented and the figures (Tables, Images) are of sufficient quality for clarity

Reviewer #2: (No Response)

**Conclusions**

-Are the conclusions supported by the data presented?

-Are the limitations of analysis clearly described?

-Do the authors discuss how these data can be helpful to advance our understanding of the topic under study?

-Is public health relevance addressed?

Reviewer #1: Yes, the conclusions are supported by the data presented

Reviewer #2: (No Response)

**Editorial and Data Presentation Modifications?**

Reviewer #1: (No Response)

Reviewer #2: (No Response)

**Summary and General Comments**

Reviewer #1: (No Response)

Reviewer #2: (No Response)

PLOS authors have the option to publish the peer review history of their article (what does this mean?). If published, this will include your full peer review and any attached files.

Reviewer #1: Yes: Nagwa Mostafa El-Sayed

Reviewer #2: No

Figure Files:

Data Requirements:

Reproducibility:

References

---

## [Editor Report · Decision Letter 1]

7 Feb 2024

Dear Dr Kazemirad,

We are pleased to inform you that your manuscript 'Chitosan nanoparticles improve the effectivity of miltefosine against Acanthamoeba' has been provisionally accepted for publication in PLOS Neglected Tropical Diseases.

Best regards,

Aiman Abu Ammar

Guest Editor

Charles Jaffe

Section Editor

---

## [Editor Report · Acceptance letter]

24 Feb 2024

Dear Dr Kazemirad,

We are delighted to inform you that your manuscript, "Chitosan nanoparticles improve the effectivity of miltefosine against *Acanthamoeba*," has been formally accepted for publication in PLOS Neglected Tropical Diseases.

Best regards,

Shaden Kamhawi

co-Editor-in-Chief

Paul Brindley

co-Editor-in-Chief
